# Wettability, Corrosion Resistance, and Osteoblast Response to Reduced Graphene Oxide on CoCr Functionalized with Hyaluronic Acid

**DOI:** 10.3390/ma15072693

**Published:** 2022-04-06

**Authors:** Belén Chico, Blanca Teresa Pérez-Maceda, Sara San-José, María Lorenza Escudero, María Cristina García-Alonso, Rosa María Lozano

**Affiliations:** 1Department of Surface Engineering, Corrosion and Durability, Centro Nacional de Investigaciones Metalúrgicas (CENIM, CSIC), Avda. Gregorio del Amo 8, 28040 Madrid, Spain; bchico@cenim.csic.es (B.C.); escudero@cenim.csic.es (M.L.E.); 2Cell-Biomaterial Recognition Lab, Department of Cell and Molecular Biology, Centro de Investigaciones Biológicas Margarita Salas (CIB-MS, CSIC), Ramiro de Maeztu 9, 28040 Madrid, Spain; sarasanj21@gmail.com

**Keywords:** CoCr alloy, hyaluronic acid, electrochemically reduced graphene oxide, wettability, osteoblast, corrosion

## Abstract

The durability of metal–metal prostheses depends on achieving a higher degree of lubrication. The beneficial effect of hyaluronic acid (HA) on the friction and wear of both natural and artificial joints has been reported. For this purpose, graphene oxide layers have been electrochemically reduced on CoCr surfaces (CoCrErGO) and subsequently functionalized with HA (CoCrErGOHA). These layers have been evaluated from the point of view of wettability and corrosion resistance in a physiological medium containing HA. The wettability was analyzed by contact angle measurements in phosphate buffer saline-hyaluronic acid (PBS-HA) solution. The corrosion behavior of functionalized CoCr surfaces was studied with electrochemical measurements. Biocompatibility, cytotoxicity, and expression of proteins related to wound healing and repair were studied in osteoblast-like MC3T3-E1 cell cultures. All of the reported results suggest that HA-functionalized CoCr surfaces, through ErGO layers in HA-containing media, exhibit higher hydrophilicity and better corrosion resistance. Related to this increase in wettability was the increase in the expressions of vimentin and ICAM-1, which favored the growth and adhesion of osteoblasts. Therefore, it is a promising material for consideration in trauma applications, with improved properties in terms of wettability for promoting the adhesion and growth of osteoblasts, which is desirable in implanted materials used for bone repair.

## 1. Introduction

The synovial fluid found in normal hip joints is an aqueous electrolyte solution generated by the cartilage and synovium, which contains as main constituents, proteins, lipids, and hyaluronic acid. Following a hip implant surgery, the synovial membrane eventually changes, generating a liquid of near composition to synovial fluid that lubricates the bearing device, known as pseudosynovial fluid [1,2]. Given the great importance of the role of the synovial fluid in the lubrication of total joint arthroplasty, it would be convenient to preserve the secretion of synovial fluid during joint replacement surgery [3].

Boundary lubrication reduces the wear of both natural and artificial joints by the adsorption of specific biomolecules of the synovial fluid, such as serum albumin, c-globulin, phospholipids, and HA [4,5], to the articulating surfaces in joints [6,7]. Hyaluronic acid, the major synovial fluid constituent in normal human joints, is responsible for the viscosity of the synovial fluid, which reaches a concentration between 3–4 mg/mL [8,9]. However, its role in the boundary lubricating ability of joints has not been clearly identified. An insight into the adsorption abilities of these molecules on sliding surfaces would be essential to reach a conclusion for understanding their lubricating mechanisms.

The effect of HA on the friction and wear of both natural and artificial joints has been reported by many studies [10,11,12]. Among them, some studies regarding macroscale frictional behavior have revealed the improvement in the lubricating effect of HA on cartilage-on-cartilage, cartilage-on-steel, and cartilage-on-glass bearings [13].

Cobalt-chrome alloys (CoCr) are commonly used for the MoM hip joints, due to their substantially low corrosion and wear rates. Through retrieval of long-lasting bearing CoCr joint prosthesis from cadavers have found that there is an improvement in friction and wear resistance when carbonaceous layers appear where the sliding in the artificial joint took place [14,15,16].

The importance of studying the surface modification with compounds that have high carbon content, such as nanocrystalline graphite or graphene-based structures, shows great promise in decreasing corrosion and enhancing the wearability of implants [17,18]. An increasing amount of GO enhances the surface properties and functionalities that encourage cell adherence, differentiation, and growth [19,20]. In addition, this has improved the physicochemical and biomechanical properties. As an example, the scaffolds sample NSC-4 with the maximum amount of GO showed considerable cell adherence with cylinder interconnected cell morphology [21].

GO contains oxygen-based functional groups that offer the possibility through H-bonding by adding other molecules, such as HA, to improve surface lubricity, given its surface hydration capacity and greater biocompatibility.

Surface modification of materials is a key property in new developments of functional implantable biomaterials used in regenerative medicine and cancer therapies. Therefore, this research aims to generate carbonaceous layers that increase the durability of metal–metal prostheses, while increasing the degree of lubrication at the same time [22]. In this paper, the main objective is the study of electrochemically reduced graphene oxide on CoCr surfaces functionalized with HA, from the point of view of wettability, corrosion resistance, and osteoblast response.

## 2. Materials and Methods

### 2.1. Materials

The biomedical grade CoCr alloy was provided by International Edge. The elemental composition (wt%) was: 27.25% Cr, 5.36% Mo, 0.69% Mn, 0.68% Si, 0.044% C, 0.02% W, 0.15% N, 0.002% Al, 0.001% S, 0.002% P, 0.002% B, 0.001% Ti, balance Co.

CoCr discs of 12 mm diameter and 2 mm thick were used. Successive abrasive papers from 600 to 2000 of silicon carbide grade were used for grinding CoCr discs. Subsequently, they were washed with distilled water.

### 2.2. Electrochemical Reduction of Graphene Oxide

Electrochemical tests were performed in an aqueous suspension of graphene oxide (4 g/L) in a three-electrode electrochemical cell. The electrochemical cell consisted of two-side CoCr discs as the working electrode, graphite as the auxiliary electrode, and Ag/AgCl as the reference electrode. The electrochemical method used to synthesize electrochemically reduced graphene oxide (ErGO) films on CoCr surfaces (namely CoCrErGO) was cyclic voltammetry (CV), from −2.1 to −0.5 V Ag/AgCl at a scanning rate of 10 mV/s for 5 cycles [18]. Raman spectroscopy and X-ray photoelectron spectrometry confirmed the presence of ErGO films with an estimation by AFM of around 27 ± 5 nm thickness on the CoCr surface [18].

### 2.3. Functionalization with Hyaluronic Acid of Electrochemically Reduced Graphene Oxide on CoCr

ErGO films deposited on the CoCr surface were washed with distilled water and further functionalized with HA (namely CoCrErGOHA) for 24 h of soaking in a PBS solution of chemical composition: 0.2 g/L KCl, 0.2 g/L KH_2_PO_4_, 8 g/L NaCl, and 1.150 g/L Na_2_HPO_4_ anhydrous, and containing 3 g/L of HA (PBS-HA). The equivalent concentrations are found in the synovial fluid [23]. The XPS spectrum (not shown) of the CoCrErGOHA surface showed a peak in binding energies corresponding to N 1s, which is associated with the presence of HA on the CoCrErGO surface.

### 2.4. Contact Angle Measurement

Contact angles (Θ) were measured using Theta (Attension) optical tensiometer (KSV Instruments, Monroe, CT, USA) with automatic multi-liquid dispenser and accompanying software. Sessile drops were recorded in fast mode with a trigger, within 5 s from the moment the drop touched the surface. Contact angles were measured using the Young–Laplace drop profile fitting method. Each contact angle value was cited as an average of three measurements performed at three different locations on the specimen surface. An average of 15 frames/s has been used to calculate the contact angle for each drop. The sessile drop volume used for the measurements was 2 μL. Wettability was derived from the drop-angle of each surface (CoCrErGOHA, CoCrErGO, and CoCr) in different media (water and PBS-HA).

#### Statistical Analysis

Quantitative variables are summarized as the number of samples (N), as well as the mean and standard deviation (SD). The influence of surface and medium on the contact angle was studied. After checking for normality and homoscedasticity assumptions, a two-way ANOVA with an independent variable, i.e., angle, and two dependent variables, surface with levels: CoCrErGOHA, CoCrErGO, and CoCr as well as medium with levels: Water and PBS-HA, was developed. The main effects of surface and medium, using Tukey’s test for multiple pairwise comparison and the surface–medium interaction effect, were studied. Moreover, the planned simple effects of the medium at each level of the surface were considered, setting the significance at 0.017 (Bonferroni correction).

*p*-values < 0.05 were considered as significant, except for the simple effect analysis. Statistical analyses were developed with IBM SPSS 23.0 (Armonk, NY, USA).

### 2.5. Electrochemical Behavior of Functionalized CoCr Surfaces

The corrosion behavior of the CoCrErGOHA surface was electrochemically studied in PBS-HA solution and assessed by electrochemical impedance spectroscopy (EIS) and anodic polarization curves (APC). The impedance data were obtained by applying a sinusoidal wave of 10 mV in amplitude versus the corrosion potential in a frequency range from 100 kHz to 1 mHz, logarithmically spaced 5 points/decade. At the end of EIS measurements (7 days), the APC were drawn at a scanning rate of 10 mV/min from corrosion potential, E_corr_, to 1.5 V polarization. In addition, the return curves were obtained until achieving the new E_corr_. The electrochemical measurements were performed at least in triplicate, each one with a working area of 78.5 mm^2^.

The impedance response was analyzed by fitting the experimental data with the adequate equivalent circuit model. The equivalent circuit elements were calculated using a non-linear least-squares program (NLLS program) included in Z-view software for all of the CoCr surfaces. The quality of the fitting was assessed by the lowest Chi-square value and relative errors (in %) for all of the fitted values.

### 2.6. Osteoblast Cell Cultures Assays

Biocompatibility tests of CoCrErGOHA, CoCrErGO, and CoCr surfaces (discs 12 mm diameter) were carried out in osteoblast-like MC3T3-E1 cell cultures, obtained from DSMZ Human and Animal Cell Bank (DSMZ, Braunschweig, Germany). Previously, both sides of the metallic discs were sterilized for 5 min under UV, in active vertical flux cabin.

The effect of the modified surfaces on cell cultures proliferation and cytotoxicity was analyzed by seeding osteoblasts cells at 125,000 cells/mL on a 24-well culture plate. A volume of 1 mL in Dulbecco’s Modified Eagle Medium (DMEM 41966; Gibco ThermoFisher Scientific, Waltham, MA, USA) supplemented with 10% heat-inactivated fetal bovine serum (FBS; Gibco ThermoFisher Scientific, Waltham, MA, USA) and a mixture of antibiotics (penicillin at 100 units/mL and streptomycin at 100 μg/mL, Gibco, BRL) were used as the culture medium. Cell cultures were maintained for 72 h in a cell culture chamber at 37 °C and 5% CO_2_. Incubation time was selected based on the previous studies [24,25]. Mitochondrial activity (WST-1 assay) and plasma membrane damage (LDH assay) were used to evaluate the biocompatibility and cytotoxicity, respectively, as described in [26].

#### 2.6.1. Measurement of Mitochondrial Activity

Reduction of the WST-1 reagent [4-(3-4-iodophenyl)-2-(4-nitro-phenyl) -2H-5-tetrazolio]-1,3-benzene disulfonate (Roche Diagnostics GmbH, Mannheim, Germany)) was used to evaluate the effect of modified surfaces on the mitochondrial activity of osteoblast cell cultures. The mitochondrial activity measurement was directly proportional to the number of metabolically active cells in culture [24]. Following the culture, 100 µL of the cell proliferation kit reagent WST-1 was added to each well, containing 1 mL of fresh complete cell culture medium and incubated inside the cell culture incubator for 180 min. Following incubation, 100 µL of each reaction mixture was transferred to a 96-well cell plate. The absorbance of the samples was measured as differential absorbance, 415 nm minus 655 nm, in an iMark microplate absorbance reader (Bio-Rad, Richmond, CA, USA). The absorbance given by the complete cell culture medium was used as a blank. Independent experiments were carried out in triplicate.

#### 2.6.2. Measurement of Lactate Dehydrogenase Activity 

Lactate dehydrogenase (LDH) activity was measured in the supernatant of cell cultures to measure and quantify the effect of modified materials on cell death and lysis. Cytotoxicity Detection Kit^plus^ (Roche Diagnostics GmbH, Mannheim, Germany) was used to measure LDH. Supernatants were collected from the cell culture after 72 h and centrifuged for 5 min at 1024 g. Complete cell culture medium was used as a control for the absorbance baseline. LDH activity was measured based on differential absorbance, 490 nm minus 655 nm, in an iMark microplate absorbance reader (Bio-Rad, Richmond, CA, USA). Quantification of LDH activity is used as an indicator of plasma membrane damage, as it is a stable cytoplasmic enzyme present in all of the cells. In addition, it is rapidly released into the cell culture supernatant when the plasma membrane is damaged, which is a sign of cell death [26]. Independent experiments were carried out in triplicate. Complete cell lysis of osteoblast cell cultures, in the presence of the materials, was carried out to determine the total LDH activity that was used here as an indirect method for quantifying the total number of osteoblasts in the culture.

#### 2.6.3. Immunofluorescence Detection of Vimentin and ICAM-1 in MC3T3-E1 Osteoblast Cell Cultures by Confocal Microscopy

Vimentin and ICAM-1 immunodetection assays were performed in osteoblast cell cultures in the presence of discs of CoCrErGOHA and CoCr for 72 h. Following this step, osteoblasts were fixed with cool methanol and incubated with 2% fetal bovine serum in PBS (FBS-PBS) to block non-specific binding of the primary antibody. A 1:1000 dilution of anti-vimentin antibody (mouse monoclonal anti-mouse; Cat. No. V5255; Sigma-Aldrich, St. Louis, MO, USA) in 2% FBS-PBS was used as the primary antibody and incubated for 1 h at room temperature. Following three wash steps with PBS for 5 min, a 1:400 dilution of Alexa Fluor^®^ 488 dye anti-mouse (Cat. No. A11001; Molecular Probes, Carlsbad, CA, USA) in a solution of 2% FBS-PBS was used as the secondary antibody and incubated for 1 h at room temperature in the dark. A 1:200 dilution of anti-ICAM-1 antibody (ICAM-1 (G-5), a mouse monoclonal antibody conjugated to Alexa Fluor^®^ 546 (sc-8439 AF546, Santa Cruz Biotech, Dallas, TX, USA) in 2% FBS-PBS was used to detect ICAM-1 and incubated for 1 h at room temperature. Following three wash steps with PBS for 5 min, the PBS was removed, and 1 mL of sterilized distilled water was added. Following staining, the cells fluorescence on surface discs was examined by a LEICA SP5 in a Confocal Scanning Microscopy equipped with a 63× objective (Confocal CIB facility, Madrid, Spain). The fluorescence of Alexa Fluor^®^ 488 dye was excited at 488 nm, measured between 500–548 nm, and visualized by confocal microscopy as green fluorescence for vimentin detection. The fluorescence of Alexa Fluor^®^ 546 dye was excited at 543 nm, measured between 550–610 nm, and visualized by confocal microscopy images as red fluorescence for ICAM-1 detection.

#### 2.6.4. Statistical Analysis of the WST-1 and LDH Assays

Differences between groups were tested with a one-way analysis of variance. In the cases where differences were significant (*p*-value ≤ 0.05), mean pairwise comparisons were computed with Tukey’s test. All of the analyses were performed with the R software version 3.6.3 (R Core Team, 2020) [27].

## 3. Results and Discussion

### 3.1. Wettability of Functionalized CoCr Surfaces

The hydrophilic character of the CoCr surfaces functionalized with ErGOHA and ErGO, with respect to CoCr, has been evaluated. Figure 1 shows the contact angle images for the three surfaces tested in the two media, water and PBS-HA.

A statistical analysis of the contact angle has been carried out, combining the variables of surface and medium composition. Table 1 shows the statistical summary of the contact angle, considering the two variables under study.

The ANOVA indicates that there is a main effect of the surface F_468,2_ = 43.764; *p* < 0.001, of the medium F_462,1_ = 17.485; *p* < 0.001, and the surface–medium interaction, F_462,2_ = 14.389; *p* < 0.001. In the post-hoc analysis using the Bonferroni correction, the effect of the surface factor is as follows: There were significant differences between CoCr and CoCrErGO, between CoCr and CoCrErGOHA, and between CoCrErGO and CoCrErGOHA (with *p* < 0.001 for the three pairs under consideration, respectively). The contact angle value is less on CoCrErGO than on CoCr, and smaller than ErGO on CoCrErGOHA. The functionalized surfaces, CoCrErGOHA and CoCrErGO, reduce the contact angle, which increases its hydrophilic character. Therefore, they are able to improve wear resistance and biocompatibility.

This confirms that not only is the surface modification of the CoCr important, but also the medium where the surface friction is carried out. For this reason, contact angle measurements were carried out in water, but also in PBS-HA solution. Considering the main effect of the medium factor, it can also be globally observed that the angle is smaller in PBS-HA than in water (significant differences between water and PBS-HA, *p* < 0.001) (Table 1).

The interaction between the surface and medium variables on the contact angle is also statistically significant. The surface–medium interaction is sustained as follows: Effect of the medium on CoCr is higher than the effect of the medium on CoCrErGO (*p* = 0.001), as well as the effect of the medium on CoCrErGO is different from the effect of the medium on CoCrErGOHA (*p* < 0.001). Significant differences between the effect of the medium on CoCr and CoCrErGOHA were not found (*p* = 0.086). Furthermore, the simple effect analysis of some components affects these results. The difference in the contact angle between the water and PBS-HA solution is statistically significant for the CoCr and CoCrErGOHA surfaces, *p* < 0.001, but not for the CoCrErGO surface, *p* = 0.087.

The decrease in the contact angle for the CoCr surface when it presents an ErGO layer on its surface is due to the stable carbon molecules adhered to the CoCr surface. These nanosheets present a low surface energy that favors sliding and exhibits a reduction in friction [28]. Graphene or its derivatives mainly possesses a lamellar structure, which facilitates sliding between the layers that reduce friction and wear of interacting surfaces under relative sliding motions. The addition of HA to the surface of ErGO provides a negative surface charge and high capacity for interaction with water [29]. The decrease in the contact angle by the presence of HA is probably due to its three-dimensional conformation in the form of a long chain arranged in bundles, creating a flexible network of interconnected molecules. Furthermore, HA creates an aqueous space inside. It is expected that this brush-shaped chain aligned along the surface is capable of cushioning the wear phenomena produced by the mechanical pressures, which is induced by the movement, due to the variable presence of water. However, as the structure of HA is constantly changing, it is difficult to know its mechanism of action.

The greater hydrophilicity found in the functionalized surface ErGOHA on CoCr in the PBS-HA medium led us to investigate the behavior of this modified surface, according to two new variables: Its resistance to corrosion in the test medium with PBS-HA and the effect this modification has on the adhesion and growth of osteoblasts.

### 3.2. Corrosion Resistance of Functionalized CoCr Surfaces in PBS-HA Medium 

Figure 2 shows the Nyquist (Z″/Z′) (a) and Bode (b) diagrams (impedance modulus, left axis, and phase angle, right axis, versus frequency) for CoCrErGOHA surfaces in PBS-HA medium with the immersion time.

Nyquist diagram shows semicircular arcs with greater amplitude as the immersion time in PBS-HA increases. This corresponds to the higher impedance modulus at lower frequencies and larger phase angles in Bode diagram (Figure 2b). This good behavior of corrosion resistance of CoCrErGOHA surfaces over time could be due to the increased hydration capture by the HA molecules, with the better wettability verified by the lowest contact angles measured (see Table 1 and Figure 1). 

The experimental impedance data have been properly fitted to two equivalent electrical circuits: Considering two time constants, in the case of CoCr surfaces functionalized with ErGOHA and ErGO (Figure 3a), and one single time constant, in the case of CoCr surface (Figure 3b).

In the circuit of Figure 3a, a time constant, R_coat_ and CPE2, could be due to the presence of HA on the ErGO surface and attributed to the processes and phenomena that take place through the adsorbed hyaluronic acid, as previously observed in the PBS media of this surface [22]. The other time constant, R_p_ and CPE1, could be due to a charge transfer process of CoCr passive film/PBS-HA medium interface that takes place through the discontinuities or coating defects of functionalized layers (Figure 3a) or directly on the passive film CoCr surface without functionalization (Figure 3b). R_e_ corresponds to the resistance of the electrolyte.

Table 2 shows the evolution over the testing time of these parameters for the CoCr surface functionalized with ErGOHA in PBS-HA medium. For comparative purposes, these parameters are also displayed after 3 days of testing for the surface modified only with ErGO and for the CoCr without functionalization. Specifically, this testing time is the same as the time used in biocompatibility studies (72 h).

The CoCrErGOHA surface presents constant R_e_ values and slightly increased R_coat_ values over the immersion time. The quasi-capacitance for n close to 1 (CPE2)and n_2_ values are similar for both layers functionalized with ErGOHA and ErGO at 3 days of immersion. 

Table 2 shows an initial n_1_ value of 0.47 for the CoCrErGOHA surface, which indicates diffusion-controlled processes. Following 3 days of immersion, this value is 0.79 (similar to the CoCrErGO surface) and increases up to 0.81 at 7 days of immersion. These changing values of n_1_ show how the system is initially controlled by diffusion to progressively be controlled by charge transfer processes, taking n_1_ values of pseudo-capacity. This indicates the movement of the electrolyte through the functionalized layer to the CoCr surface as the testing time increases.

The corrosion behavior of the two surfaces, CoCrErGOHA and CoCrErGO, in PBS-HA medium is similar both from the point of view of mechanism and corrosion resistance. However, a slight improvement in all of the parameters (Table 2), for the same testing time, is observed when the ErGO layer is additionally functionalized with HA.

Figure 4 shows the anodic polarization curves (APC) of CoCrErGOHA, CoCrErGO, and CoCr surfaces in PBS-HA. All of the surfaces show similar susceptibility to the pitting corrosion: Corrosion potential values, E_corr_, begin around 0 V; passivation intensity value, I_p_, begins at 10^−8^ A and progressively increases up to 10^−4^ A; and the repassivation potential values, E_rp_, are achieved around 0.8 V. The CoCr surfaces, whether functionalized or not, show the same susceptibility to localized corrosion, in the PBS-HA medium. 

In summary, from the contact angles and corrosion behavior, the CoCrErGOHA surface provides great hydration capabilities and does not change the corrosion resistance in PBS-HA medium. Both properties are potential characteristics for resisting the mechanical stress, but can also induce other desirable functions related to cell growth and cell adhesion, processes which are important for its application as a material for bone repair.

### 3.3. Measurement of LDH Activity in Supernatants and Cell Cultures and of MC3T3-E1 Osteoblasts Exposed to Functionalized CoCr Surfaces

Figure 5 shows the LDH content in supernatants of osteoblast-like MC3T3-E1 cell cultures. As observed, none of the assayed materials produced any significant damage on the plasma membrane, since LDH levels in the cell culture supernatants are similar in all of the tested samples. The different assayed metals produce an LDH activity similar to the control sample, an activity that corresponds to the release of the enzyme induced by the assay conditions and the time used in culture.

The total LDH content was used as an indirect measurement to quantify the total number of cells in the cultures exposed to materials. As cell adhesion is altered by the interaction with some metallic materials, as previously reported for certain magnesium alloys [30], the trypsin incubation used in routine lab cell culture to quantify the cell number does not work as an adhesion to the material, which is extremely strong.

In Figure 6, the total LDH content of osteoblast-like MC3T3-E1 cell cultures in contact with ErGOHA, ErGO, and CoCr surfaces with comparable values in CoCrErGOHA and CoCrErGO were observed, indicating that the number of osteoblasts cells is similar in these materials.

### 3.4. MC3T3-E1 Osteoblasts Biocompatibility of Functionalized CoCr Surfaces

The biocompatibility of osteoblasts was comparatively analyzed in cell cultures exposed for 72 h to CoCr, CoCrErGO, and CoCrErGOHA and in the absence of any material (control) (Figure 7). Osteoblasts mitochondrial activity, represented here by the increase in the absorbance value, is comparable in cultures exposed to CoCr, functionalized CoCr, and control assays. The exposure of cells to CoCrErGO produces an increase in the respiratory activity, although this did not reach a statistical significance for the values shown by CoCr and the control sample. However, when the osteoblasts were exposed to CoCrErGOHA, a higher respiratory activity with a statistical difference with respect to the activity seen in CoCr was observed, which can be interpreted as higher osteoblast proliferation on this surface. However, as the number of osteoblasts cells was not higher for CoCrErGOHA (Figure 6), an explanation for the increase in the respiratory activity induced by this material (Figure 7) could be associated with an oxidative stress, which is produced by the interaction with the modified metal surface.

### 3.5. Immunodetection of Vimentin and ICAM-1 in MC3T3-E1 Osteoblast Cell Cultures Exposed to CoCrErGOHA

As vimentin expression is positively regulated by the reactive oxygen species (ROS) [31], vimentin detection was comparatively analyzed in osteoblast cell cultures exposed to CoCrErGOHA and CoCr by immunofluorescence. This allows us to see whether the increase in the respiratory rate observed in osteoblasts exposed to CoCrErGOHA is in accordance with the change in vimentin expression.

Figure 8 shows the vimentin detection analyzed by immunofluorescence (top images) in MC3T3-E1 osteoblast cell cultures exposed to CoCrErGOHA and CoCr. As shown in Figure 8 (top panels), vimentin fluorescence clearly appears to increase in response to the exposure to CoCrErGOHA (Figure 8, right panel), as a large amount of highly condensed green fluorescence was observed in the growth of osteoblasts on this surface. Although the increase in vimentin could be related to the higher ROS levels produced by the exposure to CoCrErGOHA, this effect could be interpreted to indicate that upon ErGOHA functionalization the CoCr improves its biocompatibility, as previously reported. In addition, the mitochondria show less sensitivity to ROS in cells containing vimentin than in cells devoid of vimentin [32]. This could be interpreted as the high expression of vimentin that protects the osteoblasts from ROS generation. This is an advantageous property that CoCr acquires upon ErGOHA functionalization. For a long time, ROS have been exclusively considered as damaging agents accompanying acute and chronic inflammatory processes. However, ROS are also important mediators in processes as differentiation, cell adhesion and senescence, apoptosis, hormone biosynthesis, and signal transduction [33,34]. Some reports have suggested that redox signaling may have beneficial effects on the wound microenvironment to promote wound healing [31,35].

Some research papers have reported that intermediate proteins, such as vimentin, are involved in the organization and function of cell adhesion proteins, migration, and signal transduction [36] and participate in the intracellular organelle movements. To test whether the vimentin overexpression observed upon ErGOHA functionalization of CoCr is related in some way with the effect on the adhesion of osteoblasts, a comparative analysis of ICAM-1 was performed here, as it is a key protein involved in dynamic intercellular adhesion [37]. Figure 8 (bottom panels) shows images of ICAM-1 fluorescence in osteoblast cell cultures exposed to CoCrErGOHA and CoCr. It is remarkable that the functionalization with ErGOHA of CoCr produces an enhancement in ICAM-1 fluorescence, showing a higher intensity and more condensed fluorescence, in comparison with the one observed in cultures on CoCr, where a more diffuse and attenuated signal of ICAM-1 was observed. The ICAM-1 overexpression observed in osteoblasts adhered on CoCrErGOHA indicates an effect on the enhancement of the adhesion of these cells on the metallic surface. The improvement of cell adhesion is a desirable effect on the material–cell interactions, as it is a signal of material biocompatibility that promotes cell integration [38]. Adhesion is vital in determining the early osseointegration between the bone tissue and biomaterial via regulating signaling pathways [39]. As enhancing the osteoblasts cell adhesion has become one of the prime aims when optimizing surface properties of bone biomaterials, the surface modification of CoCr by ErGOHA seems to favor osteoblast adhesion, as an increase in vimentin and ICAM-1 expression is herein reported.

## 4. Conclusions

The highest hydrophilicity occurs on the ErGOHA functionalized CoCr surface in the PBS-HA medium.The corrosion behavior of CoCrErGOHA and CoCrErGO is similar from the point of view of mechanism and resistance in the PBS-HA medium.High expressions of vimentin and ICAM-1 upon ErGOHA functionalization favor osteoblast adhesion. The increase in ICAM-1 expression together with the better cell adhesion observed in the presence of ErGOHA can be related to the increase in wettability, favoring the adhesion of cells to the surface, which is desirable in the case of implanted materials used for bone repair.Functionalization with HA through ErGO layers improves the applicability of CoCr in artificial prostheses, due to the increased wettability and cell adhesion, maintaining corrosion resistance in a physiological medium containing HA.

## Figures and Tables

**Figure 1 materials-15-02693-f001:**
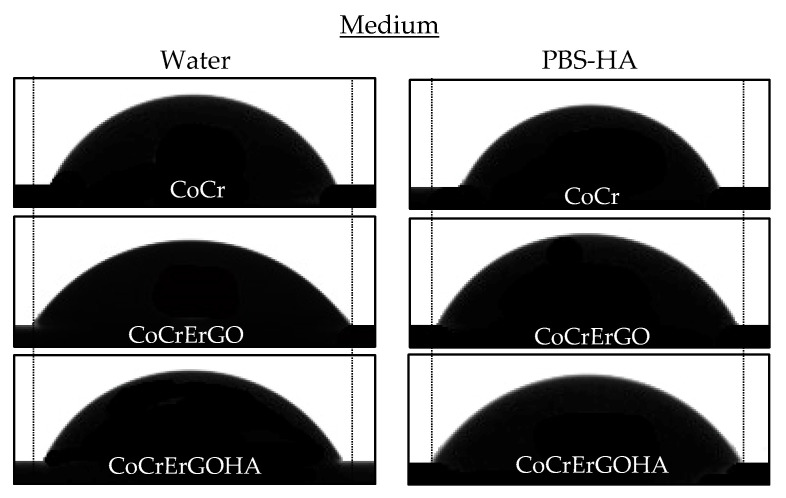
Contact angle images for CoCr, CoCrErGO, and CoCrErGOHA surfaces in water and PBS-HA.

**Figure 2 materials-15-02693-f002:**
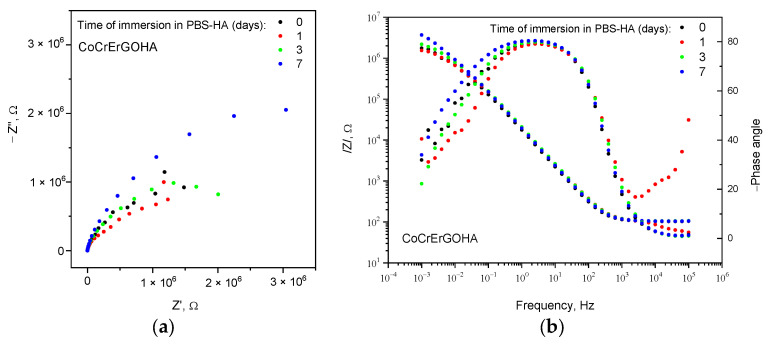
Nyquist (**a**) and Bode diagrams of impedance modulus ((**b**) left axis) and phase angle ((**b**) right axis) versus frequency for CoCrErGOHA surface in PBS-HA medium with the immersion time.

**Figure 3 materials-15-02693-f003:**
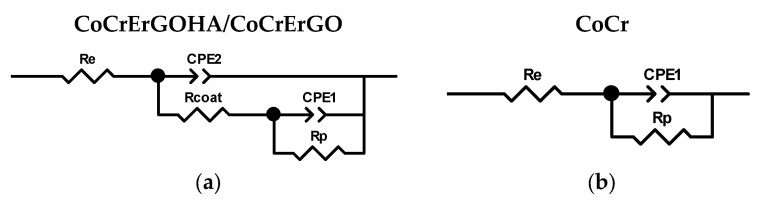
Equivalent circuits used for the fitting of the impedance response of CoCrErGOHA, CoCrErGO, and CoCr surfaces during testing times considering two constant times (**a**) and one constant time (**b**).

**Figure 4 materials-15-02693-f004:**
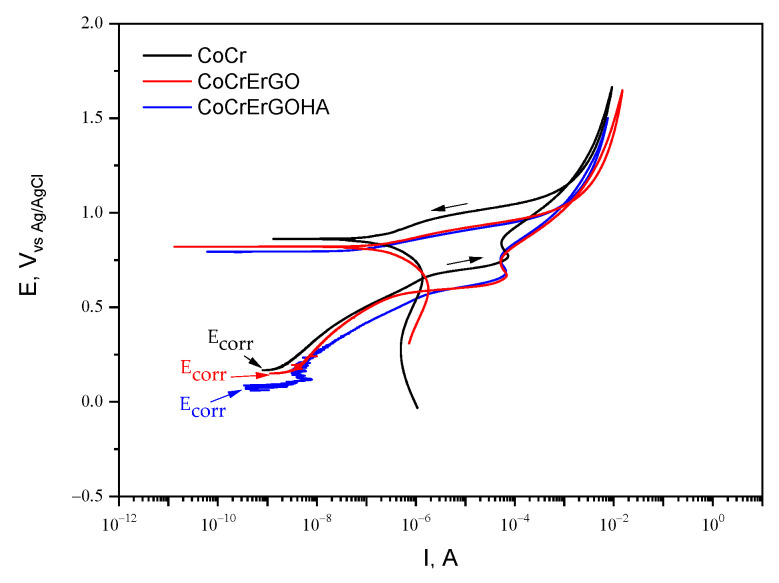
Anodic polarization curves (APC) of CoCrErGOHA, CoCrErGO, and CoCr surfaces in PBS-HA medium.

**Figure 5 materials-15-02693-f005:**
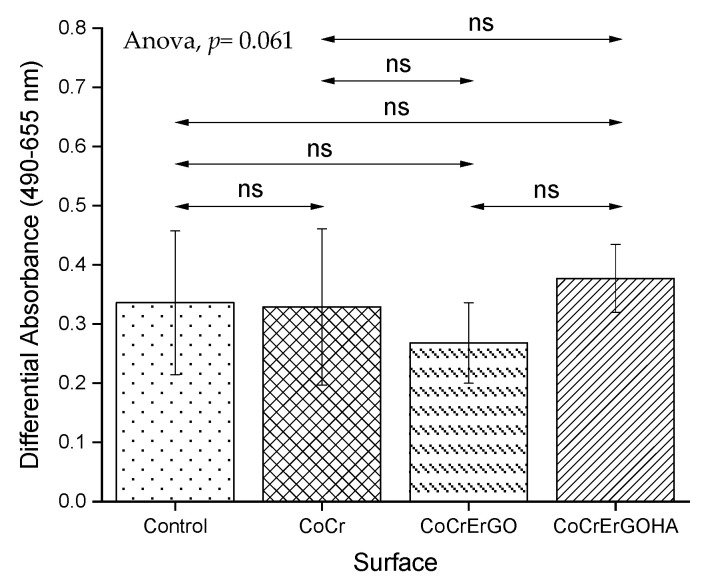
Effect of CoCr, CoCrErGO, and CoCrErGOHA on the LDH content in supernatants of osteoblast-like MC3T3-E1 cell cultures. Osteoblasts were also cultured in the absence of any discs, and used as control. Differences between the studied samples did not reach statistical significance. ns: *p* > 0.05.

**Figure 6 materials-15-02693-f006:**
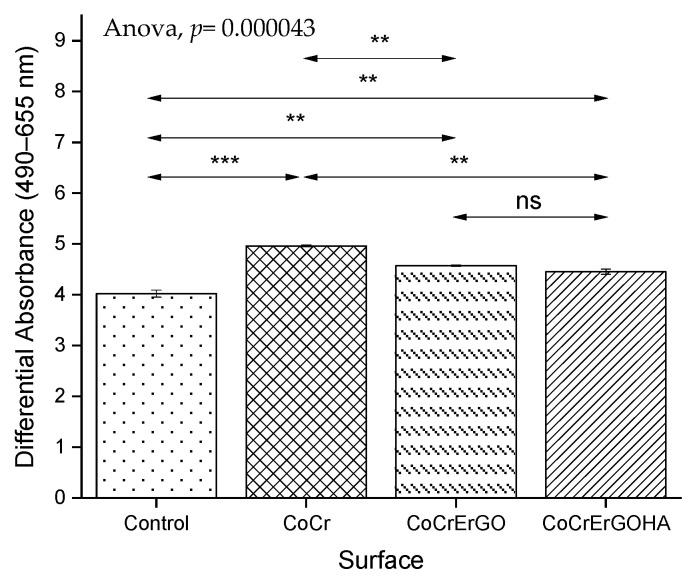
Effect of CoCr, CoCrErGO, and CoCrErGOHA on the total LDH content of osteoblast-like MC3T3-E1 cell cultures. Osteoblasts were also cultured in the absence of any discs, and used as control. ns: *p* > 0.05, **: *p* ≤ 0.01, ***: *p* ≤ 0.001.

**Figure 7 materials-15-02693-f007:**
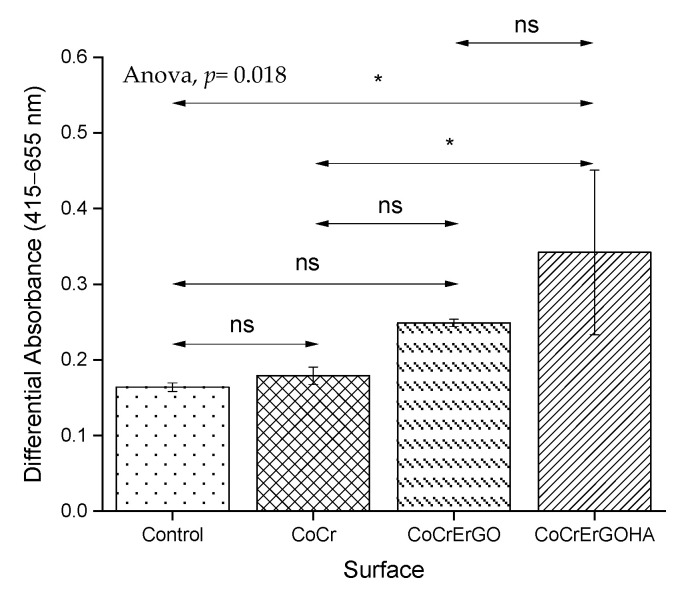
Effect of CoCr, CoCrErGO, and CoCrErGOHA on the proliferation and viability of osteoblast-like MC3T3-E1 cell cultures. Osteoblasts were also cultured in the absence of any discs, and used as control. ns: *p* > 0.05, *: *p* ≤ 0.05.

**Figure 8 materials-15-02693-f008:**
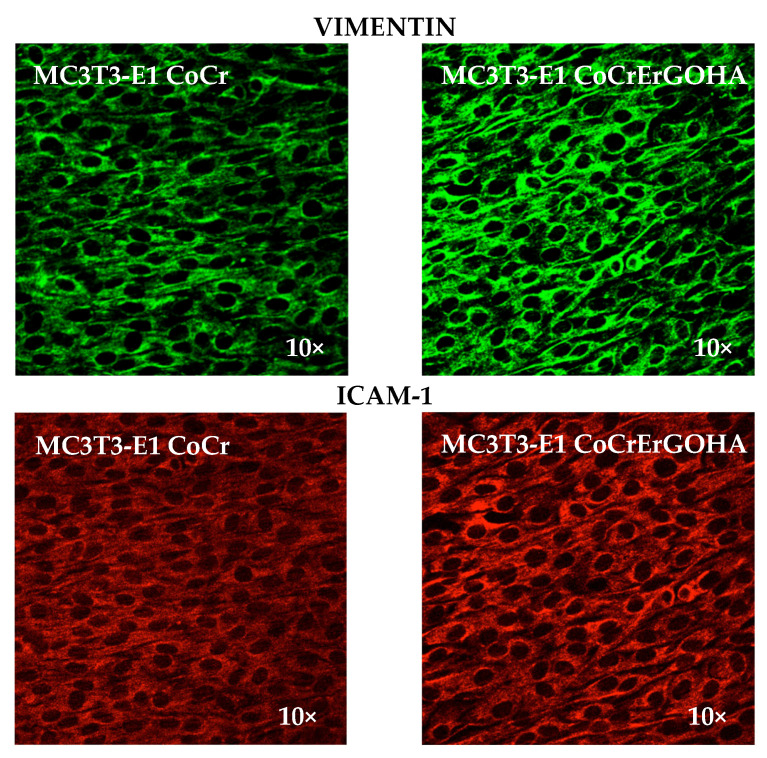
Vimentin detection analyzed by immunofluorescence (**top**) and ICAM-1 (**bottom**) fluorescence in MC3T3-E1 osteoblast cell cultures exposed to CoCrErGOHA and CoCr.

**Table 1 materials-15-02693-t001:** Statistical data of the contact angle in different surface combinations: CoCr, CoCrErGO, and CoCrErGOHA and medium: Water and PBS-HA.

	Total			Medium					
				Water			PBS-HA		
Surface	N	Mean	SD	N	Mean	SD	N	Mean	SD
CoCr	152	69.4	3.4	72	71.4	2.0	80	67.5	3.4
CoCrErGO	144	64.5	6.4	68	63.5	5.0	76	65.5	7.3
CoCrErGOHA	178	61.5	10.7	76	65.3	6.6	102	58.6	12.2
Total	474	64.9	8.4	216	66.8	6.0	258	63.4	9.7

SD: Standard deviation.

**Table 2 materials-15-02693-t002:** Fitting values of the experimental impedance data using the equivalent circuits in Figure 3. R_e_: Electrolyte resistance; CPE1: Constant phase element and R_p_: Resistance related to metal surface/electrolyte interface; CPE2: Constant phase element and R_coat_: Ionic resistance due to the films deposited on the CoCr surface.

Surface	Time, d	R_e_ ± Error %, Ω	R_coat_ ± Error %, MΩ	CPE2 ± Error %, μs^n^/Ω	n_2_ ± Error %	R_p_ ± Error %, MΩ	CPE1 ± Error %, μs^n^/Ω	n_1_ ± Error %	Chi^2^
CoCrERGOHA	0	103.6 ± 0.5	0.4 ± 29.8	10.7 ± 1.1	0.90 ± 0.2	3.94 ± 41.9	3.87 ± 32.7	0.47 ± 18.5	0.0003
	1	101.9 ± 2.1	0.6 ± 13.5	9.30 ± 2.0	0.89 ± 0.5	1.38 ± 21.7	17.1 ± 49.8	0.78 ± 14.9	0.0021
	3	105.3 ± 0.3	0.9 ± 3.4	9.11 ± 0.4	0.90 ± 0.1	1.74 ± 3.9	12.5 ± 12.0	0.79 ± 3.1	0.0001
	7	105.1 ± 0.4	1.6 ± 6.1	9.79 ± 0.5	0.90 ± 0.1	3.61 ± 5.9	7.74 ± 17.8	0.81 ± 4.5	0.0002
CoCrErGO	3	88.2 ± 0.8	0.5 ± 9.7	9.89 ± 1.2	0.89 ± 0.3	0.71 ± 10.8	14.9 ± 33.6	0.75 ± 9.2	0.0007
CoCr	3	107.8 ± 0.7	-	-	-	5.12 ± 1.6	7.15 ± 0.5	0.92 ± 0.1	0.0005

## Data Availability

Not applicable.

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
