# Peer review of "Wettability, Corrosion Resistance, and Osteoblast Response to Reduced Graphene Oxide on CoCr Functionalized with Hyaluronic Acid"

_materials, 2022, doi:10.3390/ma15072693_

Round 1

Reviewer 1 Report

In this manuscript Wettability, Corrosion Resistance and Osteoblast Response to Reduced Graphene Oxide on CoCr Functionalized with Hyaluronic Acid, Belén Chico and other authors report the graphene oxide layers reduced on CoCr surfaces (CoCrErGO) and treated with hyaluronic acid (HA). HA is considered to be beneficial on the friction and wear of both natural and artificial joints. 

The manuscript is well structured and written. The introduction is in-depth and covers a broad area of functional implantable biomaterials. The fabrication of the functionalized samples is discussed in detail. Mitochondrial and lactate dehydrogenase activities are better for CoCrErGOHA. The contact angle, corrosion resistance of CoCrErGOHA surfaces over time, osteoblasts biocompatibility, impedance data and other characteristics of the functionalized samples are shown to be improved. However, the detailed results are only compared for CoCr, CoCrErGO, CoCrErGOHA. A comparison with the other lubricating fluid in a similar way it is done on page 11. The conclusions are fully supported by results.

I can recommend the publication of this manuscript in Materials after minor corrections.

Author Response

Please, see attachment

Reviewer 2 Report

Review of paper no. materials-1661890 titled Wettability, corrosion resistance and osteoblast response to reduced graphene oxide on CoCr alloy functionalized with hyaluronic acid by B. Chico et al.

This is an interesting article that studies the corrosion resistance, wetting properties, and biological activity of CoCr alloy substrate coated with graphene oxide and functionalized with hyaluronic acid. The paper is publishable subject to revision.

1.The authors should explain the choice of the alloy. Co-based alloys are typically more expensive compared to steel. Why have you decided to study the Co-based alloy? Is it economically feasible for hip implants? Furthermore, the alloy also contains a large amount of Cr. The biologically beneficial effects of Cr are debated. The authors should explain the choice of the alloy in the Introduction.

2.The authors coated the alloy with reduced graphene oxide (GO, lines 92-98). However, there is no experimental evidence provided for GO. Please include either an FTIR or XRD patterns to confirm that you actually had the GO coating on the surface.

3.The coatings were further functionalized with hyaluronic acid (lines 101-105). However, again there is no experimental evidence of it. Please include at least an FTIR analysis of the coating.

4.How thick were the coatings/films? Please, include an SEM image of the cross section.

5.The mean contact angles are duplicated in Fig. 1 and Table 1. Please remove Table 1 and include standard deviations in Fig. 1.

6.Contact angles are nearly identical, considering the large standard deviation (Table 1, Fig. 1). You claim that the contact angles were smaller for coated materials (lines 442-443). It is highly questionable judging from Fig. 1 alone. It is recommended to include photographs of the droplets to actually see the difference between the materials.

7.The polarization curves are not analyzed (Fig. 4). It is necessary to analyze them by Tafel extrapolation and provide Ecorr and jcorr values. Corrosion parameters should be compared with previously published studies.

8.The uncoated alloy seems to have a nobler corrosion potential compared to coated materials (Fig. 4). Have you measured an open circuit potential too?

Round 2

Reviewer 2 Report

Authors answered most of my comments. The paper is publishable.